# Prediction of gastrointestinal functional state based on myoelectric recordings utilizing a deep neural network architecture

Mahmoud Elkhadrawi [1]*, Murat Akcakaya[1], Stephanie Fulton[2], Bill J. Yates[3,4], Lee E. Fisher[5,6,7], Charles C. Horn[2,4,8,9]

1 Department of Electrical and Computer Engineering, University of Pittsburgh School of Engineering, Pittsburgh, PA, United States of America, 2 UPMC Hillman Cancer Center, University of Pittsburgh School of Medicine, Pittsburgh, PA, United States of America, 3 Department of Otolaryngology, University of Pittsburgh School of Medicine, Pittsburgh, PA, United States of America, 4 Center for Neuroscience, University of Pittsburgh, Pittsburgh, PA, United States of America, 5 Rehab Neural Engineering Labs, University of Pittsburgh, Pittsburgh, PA, United States of America, 6 Department of Physical Medicine & Rehabilitation, University of Pittsburgh School of Medicine, Pittsburgh, PA, United States of America, 7 Department of Bioengineering, University of Pittsburgh School of Engineering, Pittsburgh, PA, United States of America, 8 Division of Hematology/Oncology, Department of Medicine, University of Pittsburgh School of Medicine, Pittsburgh, PA, United States of America, 9 Department of Anesthesiology and Perioperative Medicine, University of Pittsburgh School of Medicine, Pittsburgh, PA, United States of America

* mae116@pitt.edu

**Data Availability Statement:** Please find the DOI for our data https://doi.org/10.5281/zenodo.8148338. It contains the original data and the code used for preprocessing and classification.

## Abstract

Functional and motility-related gastrointestinal (GI) disorders affect nearly 40% percent of the population. Disturbances of GI myoelectric activity have been proposed to play a significant role in these disorders. A significant barrier to usage of these signals in diagnosis and treatment is the lack of consistent relationships between GI myoelectric features and function. A potential cause of this issue is the use of arbitrary classification criteria, such as percentage of power in tachygastric and bradygastric frequency bands. Here we applied automatic feature extraction using a deep neural network architecture on GI myoelectric signals from free-moving ferrets. For each animal, we recorded during baseline control and feeding conditions lasting for 1 h. Data were trained on a 1-dimensional residual convolutional network, followed by a fully connected layer, with a decision based on a sigmoidal output. For this 2-class problem, accuracy was 90%, sensitivity (feeding detection) was 90%, and specificity (baseline detection) was 89%. By comparison, approaches using hand-crafted features (e.g., SVM, random forest, and logistic regression) produced an accuracy from 54% to 82%, sensitivity from 46% to 84% and specificity from 66% to 80%. These results suggest that automatic feature extraction and deep neural networks could be useful to assess GI function for comparing baseline to an active functional GI state, such as feeding. In future testing, the current approach could be applied to determine normal and disease-related GI myoelectric patterns to diagnosis and assess patients with GI disease.

**Funding:** "Please add BJY along with CCH and LEF as a funding receiver for grant number grant 5R01DK121703. The funders had no role in study design, data collection and analysis, decision to publish, or preparation of the manuscript".

**Competing interests:** The authors have declared that no competing interests exist.

## Introduction

Functional and motility-related gastrointestinal (GI) disorders affect nearly 40% percent of the population, with annual costs of several billion USD. Myoelectric signals have the potential to contribute to clinical diagnosis and treatment of GI disorders; however, current methods of electrogastrography (EGG), based on abdominal surface recordings, and arbitrarily-defined analytic measures, such as dominant frequency, have provided limited insight into GI functional states [1–3]. This has led to little usage of GI signaling features in the diagnosis and treatment of patients.

Machine learning approaches have been used to increase the predictive power of GI signals. Human studies have used recordings of EGG signals from the abdominal skin surface, which limited signal-to-noise resolution because of the distance of electrodes from the stomach. In this approach, subjects must also remain seated and limit movements to provide a low noise signal. Using non-invasive EGG, a support vector machine (SVM) classifier performed at 88% accuracy to predict the occurrence of motion sickness in one study [4], although EGG was only one of the inputs to the model. SVM also performed well when classifying individuals with functional nausea, with an F1-score of 0.85 [5]. A neural network approach performed at 95% accuracy to predict un-fed vs. fed states in a population of 1000 human subjects [6]. Lastly, in our prior ferret study, we achieved > 75% accuracy in a binary or 3-state classification for predicting baseline and early and late periods before emesis; importantly, in our study, electrodes were placed directly on the surface of the stomach [7].

To more fully determine the clinical potential of the above approaches, we now combine three methods: (1) free-moving/awake testing, because this will permit application in diagnostic testing in different environments, e.g., clinic and at-home; (2) chronic implantation of electrodes on multiple stomach sites (four locations), to increase signal resolution and degrees of freedom in feature engineering; and (3) deep learning with neural networks. We also compare the performance of the deep learning approach with other machine learning approaches including support vector machine (SVM), discriminant analysis, logistic regression, and random forest. We used ferrets for the current study because they are a gold-standard model for GI research, including studies to determine efficacy of anti-emetic medications now used in the clinic [8, 9].

## Materials and methods

### Animals

This study included four adult purpose-bred influenza-free male ferrets (*Mustela putorius furo*; Marshall BioResources, North Rose, NY, USA; body weights 1.4 ± 0.2 kg and age of 5.1 ± 1.2 months, mean ± SD). Animals were adapted to the housing facility for at least 7 days before surgery. Ferrets were housed in wire cages (62 × 74 × 46 cm) under a 12 h standard light cycle (lights on at 0700 h), in a temperature (20–24°C) and humidity (30–70%) controlled environment. Food (ferret kibble; Mazuri Exotic Animal Nutrition, St. Louis, MO) and drinking water were freely available; however, food was removed 3 h before experimentation to assure the stomach was empty. At the end of the study, ferrets were euthanized with an intracardiac injection of euthanasia solution (390 mg/ml sodium pentobarbital; SomnaSol EUTHANASIA-III Solution, Covetrus, Dublin, Ohio, USA) under isoflurane general anesthesia (5%). The University of Pittsburgh Institutional Animal Care and Use Committee (IACUC) approved all experimental procedures.

### Chronic electrode implant surgery

Anesthesia was induced using an intramuscular injection of ketamine (20 mg/kg), all surgical sites were shaved, and animals were endotracheally intubated with a 3.0 or 3.5 French

endotracheal tube. Surgery was conducted in an aseptic surgical suite and animals were maintained under general anesthesia using isoflurane (1–2%) vaporized in $O_2$. Subcutaneous injections of sterile saline were used to replace fluid loss. Body temperature was maintained at 36–40°C using a heating pad and monitored with a rectal probe.

Surgical procedures were similar to those in our prior study [7]. Animals were implanted with four gastric electrodes on the ventral surface of the stomach, including one on the fundus, two on the body, and one on the antrum (Fig 1), as well as two intestinal electrodes, and one abdominal vagus nerve cuff electrode (animal 103–21 was not implanted with a cuff electrode). Connectors from the electrodes were placed on the skull. Data collected from the intestine and the vagus nerve electrodes are not reported because they are not relevant to the current analysis. Each gastric planar electrode was attached to the GI serosal surface using 8–0 surgical silk with eight suture locations, four around each contact point. This was followed by placing a single suture around all electrode leads to anchor them to the left abdominal muscle wall. Before closing, the abdominal cavity was flushed with Cefazolin (1g/L), an antibiotic. The abdominal muscle was closed with 4–0 resorbable suture material and the skin with 3–0 monofilament nylon suture material and followed by applying surgical glue to the incision site.

### Behavioral testing and gastric myoelectric recording

Animals were acclimated to the test chambers for three days prior to testing. GI myoelectric activity was recorded using a cable tethered to the head connector. Baseline GI myoelectric

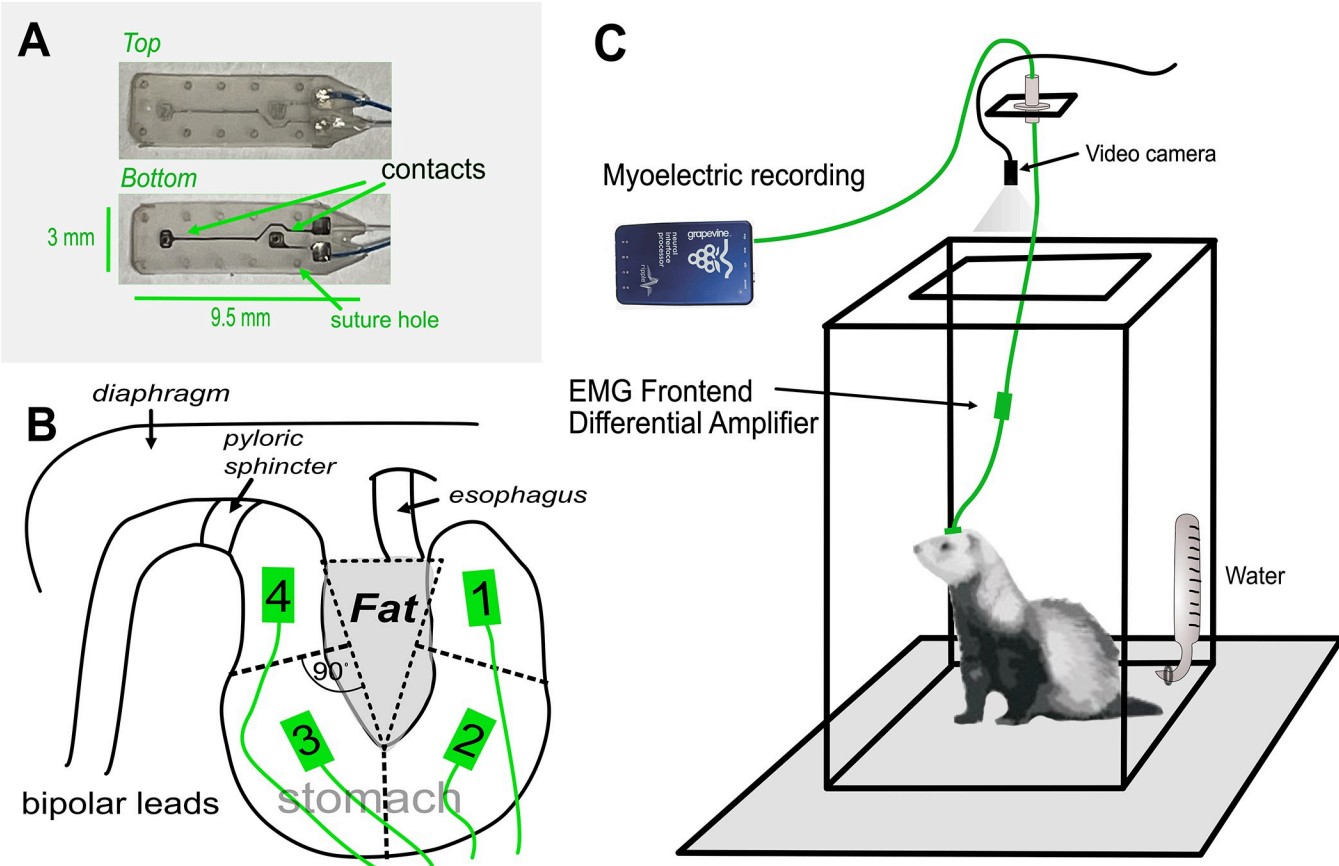

**Fig 1. Gastric myoelectric electrodes and behavioral testing.** A) Geometry of the two-contact electrodes. B) Surgical placement of the electrodes on quadrants of the ventral stomach surface of the ferret relative to the fat pad in the lesser curvature. C) Recording chamber and tether system.

activity was recorded for 1 h for the first testing session, during which no food was provided. In subsequent sessions, all animals were presented with food (Ensure Original Vanilla Flavor nutritional shake, Abbott Laboratories, Lake Bluff, Illinois, USA) for 15 min. For feeding trials, baseline myoelectric activity was recorded for 15 min prior to food presentation and 30 min after food removal. There were 1 to 2 days between each test. Myoelectric activity was recorded with differential amplifiers attached to a Ripple Grapevine acquisition system (EMG headstage, Ripple Neuro LLC, Salt Lake City, UT). Data were acquired at 2 KHz sample rate using a band pass of DC to 2 KHz. The input range was ±187.5 mV to permit large signal changes due to potential movement artifacts.

## Data analysis

**Preprocessing.** Data were initially imported into Spike 2 software (Version 9; Cambridge Electronic Design Ltd, Cambridge, England) to assess data quality and then files were exported in CSV file format. A custom workflow in Python 3 was written to process these data prior to application of machine learning algorithms, including: (1) down sampling; (2) removing dropped/saturated signals; (3) applying a band-pass filter; (4) removing movement artifacts; and (5) assessing signal quality using power spectral density (PSD) and spectrogram plots. Down sampling to 200 Hz sampling rate was accomplished with an anti-aliasing filter (https://docs.scipy.org/doc/scipy/reference/generated/scipy.signal.decimate.html). Data that exceeded the input range of the amplifier were then removed and replaced with zeros. A bandpass Butterworth filter at 0.05 to 0.7 Hz was applied, forward and backward (https://docs.scipy.org/doc/scipy/reference/generated/scipy.signal.butter.html, https://docs.scipy.org/doc/scipy//reference/generated/scipy.signal.filtfilt.html). This was followed by linear interpolation of data that exceed the physiological range of the signals, using ± 2 mV as the threshold (https://pandas.pydata.org/pandas-docs/stable/reference/api/pandas.DataFrame.interpolate.html). Fig 2 demonstrates the quality of our data with examples of raw and processed signal. A GitHub repository is available for this preprocessing pipeline. For machine learning, we only included trials that included < 20% removal of data based on dropped signals, < 20% data interpolated based on movement artifacts, and > 1000 $V^2$/Hz amplitude in the total PSD.

**Machine learning.** In this study, we were interested in classifying EGG signals between feeding and baseline control. To achieve this, we applied a deep neural network approach for automatic feature extraction. We also compared the neural network approach to other common machine learning approaches, including support vector machine (SVM), discriminant analysis (DA), logistic regression (LR), and random forest (RF) classification, which use hand-crafted features. We used Pytorch (v 1.12.1, CUDA 11.6) on Python (v 3.8.2) for the deep learning approach, and MATLAB (R2021a) for the other techniques. For all approaches, 1-hour data from each of the 4 animals was segmented into 1-min windows, thus obtaining a total of 480 data samples (240 feeding, 240 baseline). All approaches were validated using 3-fold cross validation.

Support Vector Machines (SVMs) are a type of supervised machine learning algorithm used for classification and regression analysis. They find a hyperplane in a high-dimensional space that maximally separates different classes of data or approximates the relationship between input and output variables (here we use it to classify). SVMs are useful for non-linearly separable data as they can use a kernel function to transform input data non-linearly into a different high-dimensional space like the radial basis function (RBF) kernel. Discriminant Analysis is another supervised machine learning algorithm used for classification. There are two main types of discriminant analysis: Linear Discriminant Analysis (LDA) and Quadratic Discriminant Analysis (QDA). LDA assumes that the covariance matrix of the input data is

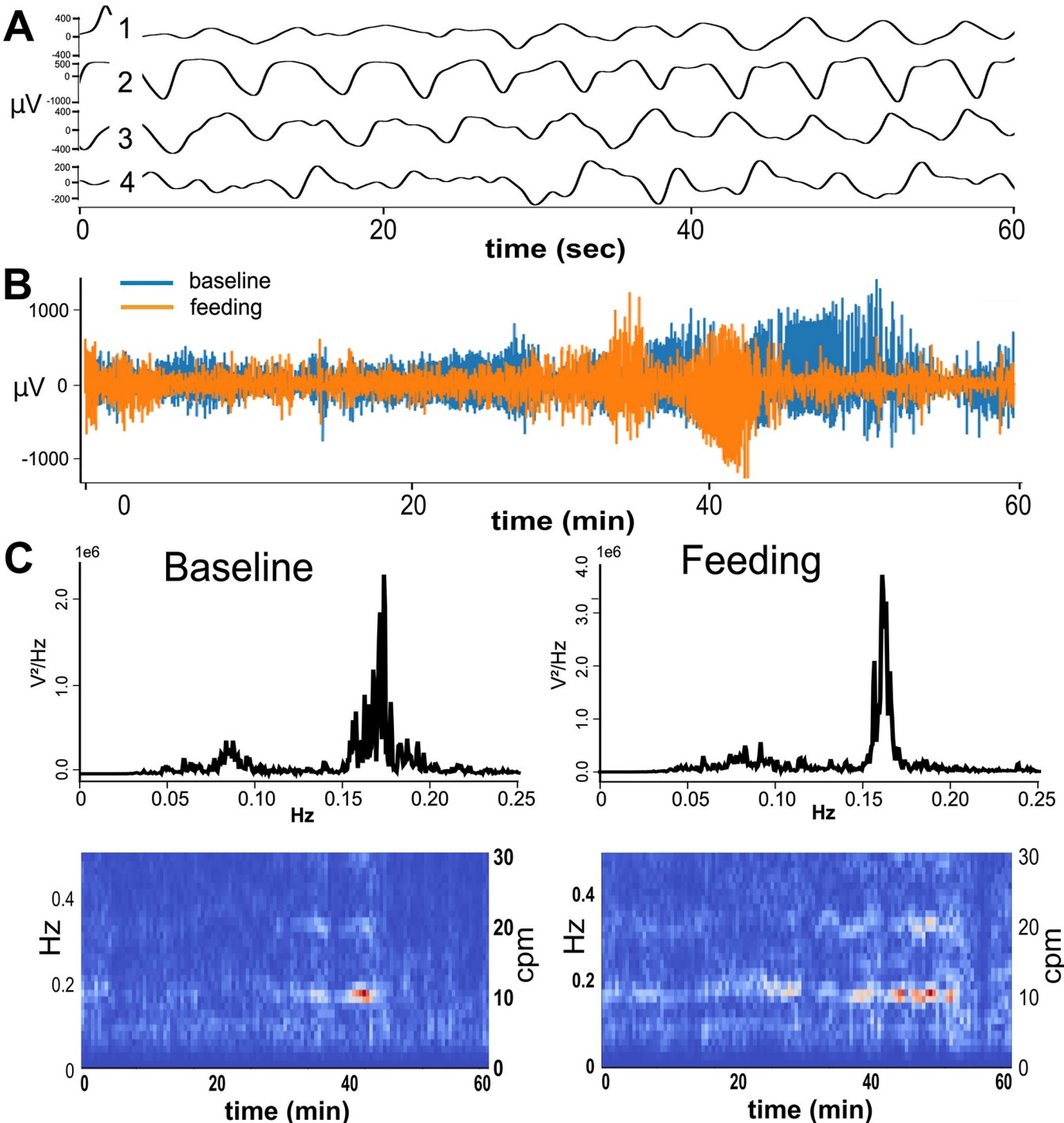

**Fig 2. Representative samples of myoelectric signals.** A) Slow wave signals from the four gastric myoelectric channels after preprocessing. B) Full length of the recorded signals for baseline and feeding conditions after preprocessing. C) Power spectral density (PSD) and spectrogram plots of the slow wave signals from channel 4, left = baseline and right = feeding. The Y-axis of the PSD plots is in E (scientific) notation. cpm = cycles per minute.

the same for all classes and that the decision boundary between classes is linear. QDA, on the other hand, assumes that each class has its own covariance matrix and that the decision boundary between classes is quadratic. Logistic Regression is a supervised machine learning

algorithm used for binary classification, where the goal is to predict the probability of an input belonging to one of two possible classes. It uses the logistic function (sigmoid function) to transform the output of a linear regression model into a probability score between 0 and 1. Random Forests are another type of machine learning algorithm used for classification and regression. They combine multiple decision trees to improve accuracy and reduce overfitting. To reduce the risk of overfitting, each decision tree is trained using a bootstrap sample of the input data and a random subset of features. Additional details about these methods can be found elsewhere [10].

For deep learning, the network was a 1D residual convolutional neural network (CNN) to extract features from the channels of gastric myoelectric signals (Fig 3). A 1D CNN is a type of neural network architecture commonly used for processing and analyzing one-dimensional signals, such as audio and time-series data [11–13]. In a 1D CNN, input data are processed through a series of convolutional layers, which learn to extract relevant features from the input signal. Each convolutional layer consists of a set of filters, which slide over the input data and perform element-wise multiplication and sum operations to produce a feature map. The resulting feature maps are then passed through a nonlinear activation function, such as a rectified linear unit (ReLU) to introduce nonlinearity into the model. These features are then passed to a fully connected neural network followed by a sigmoid activation function, that outputs a value between zero and one to classify the input signal. A residual neural network includes a layer where the output of a layer is summed with the input to that layer, effectively creating a "shortcut" connection that bypasses one or more layers. This allows the network to learn residual functions, which are easier to optimize than the original functions [15].

The input myoelectric signal was one minute long and had a sample rate of 200 Hz, containing 12,000 data points. The CNN included 6 layers, and each layer downsampled the signal by a factor of 2, except for the sixth layer which was a convolutional layer with kernel size of one, followed by a batch normalization layer. This last layer is used to achieve an output of only one channel, which corresponded to the features that were passed to a fully connected layer. After these down-sampling procedures, the original 12,000 point signal was converted to

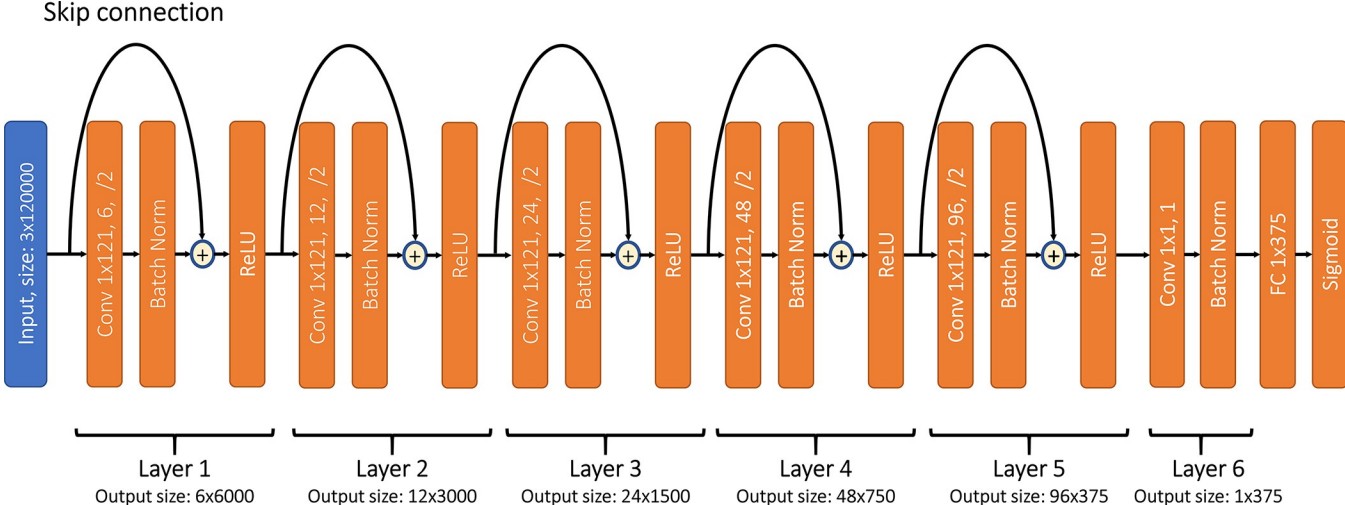

**Fig 3. Architecture of the convolutional neural network.** The convolutional layer parameters are described as follows: Conv [kernel size], [Number of output channels], [/2 means the input is downsampled by 2]. The skip connection of a block that downsamples the input is made of a convolutional layer with kernel size of 1 and stride of 2 followed by a batch normalization layer. The convolutional network is then followed by a fully connected layer, then a sigmoid activation function.

a vector of output features with a length of 375. The number of output channels for the first five layers was 6, 12, 24, 48 and 96 successively. Each layer was made of one residual block, such that each block was a convolutional layer (kernel size = 121) followed by a batch normalization layer. Then the input of the block was summed with the output of its batch normalization layer using a skip connection, and the output was passed to a ReLU activation function. The skip connection downsampled the input for the residual block to achieve a convolutional layer of stride equal to 2 and a kernel size equal to 1. The features were then fed to a fully connected layer, which passed its output to a sigmoid activation function with an output range of 0 to 1, which correspond to posterior probabilities, with 0 corresponding to the baseline and 1 corresponding to feeding. The loss function used to train this network was a binary cross-entropy loss. The performance was evaluated using ROC curves. The hyperparameters of the network (number of layers, kernel sizes, etc.) were chosen heuristically by trial and error.

In order to reduce the generalization error, we used the dropout technique [14] after each convolutional layer during training. The network was trained using early stopping to reduce overfitting [15]. Early stopping saves the model that achieves the lowest validation loss and stops the training of the network if it cannot achieve a lower validation loss during the following 50 epochs. The maximum number of training epochs was set to 200. The training batch size was set to 20 with a learning rate of 0.001. The Adam optimizer with weight decay (AdamW) was used [16].

For other classification approaches (i.e., SVM, DA, DA, and RF), we extracted frequency-based features (i.e., the signal power from narrow frequency bands) from each of the four channels of 1-min signals. We calculated the PSD for each channel using Welch's method [17] (using 16384($2^{14}$) FFT points). The PSD was normalized (dividing by total power) for each channel and for each animal individually. Prior work from our lab and others has shown that gastric slow wave activity in the ferret typically spans 0.05 to 0.3 Hz, and that activities such as feeding and emesis can increase signal power in the tachygastric (i.e., higher frequency) range of the power spectrum [7, 18]. As such, we included PSD values ranging from 0.05 to 0.7 Hz. To capture variability in different frequency bands across this range, we divided the PSD of each channel into 10 bands and summed the PSD bins for each band. Thus, we obtained 30 features from each of the 1-min 4-channel samples. We applied forward and backward sequential feature selection [19] to select the most relevant features. The feature selection algorithm minimized the misclassification rate as an objective function and performed 3-fold cross-validation with 100 Monte-Carlo simulations to choose the best features. The performance of the selected features was measured performing 3-fold cross-validation, then plotting ROC curves and computing area under the curve (AUC) values. Note that during cross-validation, the training sets features were standardized using z-scores. The computed mean and standard deviation for each training set were then used to standardize the corresponding validation set. Like the NN approach, the output from all the models ranges from 0 to 1, with 0 corresponding to the baseline and 1 corresponding to feeding, except for the SVM models, where the output score values $\in \mathbb{R}$. Thus, posterior probabilities were fitted on the SVM scores to produce output values of range 0 to 1.

## Results

As mentioned above, we collected 1 hour of data with sampling frequency of 2000 Hz (downsampled to 200 Hz) from each animal, for both baseline and feeding conditions. We segmented those signals into smaller 1-minute trials to generate multiple samples for each condition, which enabled us to train the CNN and other machine learning models to classify between feeding and baseline conditions. We compared the performance of the machine

learning methods for this classification. Note that our method did not investigate the changes across baseline and feeding based of the short signal duration during this period.

## Preprocessing

Gastric channels 1, 2, and 4 from the four animals had < 20% dropped signals, < 20% artifacts removed, and power > 1000 V2/Hz in the frequency range of interest (Table 1). In contrast, channel 3 from two of the animals did not achieve these criteria (Table 1); therefore, our machine learning predictive models were focused on using gastric channels 1, 2, and 4.

## Machine learning

All approaches were validated using 3-fold cross-validation, and the prediction scores were used to plot ROC (receiver operating characteristic) curves and calculate the area under the curve (AUC). In Fig 4A and 4B, we show the results for the methods that use hand-crafted frequency-based features, where we used forward feature selection (Fig 4A) and backward feature selection (Fig 4B). For both, we observe that the SVM model with Radial Basis Function (RBF) kernel achieved the highest performance, with AUC values of 0.88 and 0.87 for forward and backward feature selection respectively. The average AUC value for the forward feature selection approaches was 0.77, which is lower than the average AUC of the backward feature selection methods, which was 0.82.

In Fig 4C, we show the results for the deep learning approach. It outperformed the best frequency-based approach, with an AUC value of 0.96.

To characterize the variability of the performance of the deep learning approach, we repeated the 3-fold cross-validation process 10 times, with a detection threshold of 0.5. This threshold was selected because the classifier outcomes were transformed to represent probabilities from 0 to 1. Then, we calculated the average accuracy, sensitivity, and specificity and their standard deviation, which were 90.4% (1.7%), 89.4% (2.7%) and 91.3% (1.4%) respectively. For comparison, Fig 5A shows accuracy, sensitivity, and specificity for SVM, LDA, QDA, LR, and RF classifiers using 3-fold cross-validation. Various frequency features contributed to the overall performance of SVM, LDA, QDA, LR, and RF classifiers, with no specific pattern observed (Fig 5B). Fig 5B shows the contribution of each frequency bin (power) for each classifier. The

**Table 1. Assessment of channel quality.**

| channel | % signal dropped | | | | % artifact removed | | | | Power amplitude (V²/Hz) | | | |
|---|---|---|---|---|---|---|---|---|---|---|---|---|
| | 1 | 2 | 3 | 4 | 1 | 2 | 3 | 4 | 1 | 2 | 3 | 4 |
| 58–21 | | | | | | | | | | | | |
| baseline | 0.5 | 0.5 | 0.5 | 0.5 | 2.0 | 6.1 | 1.9 | 1.2 | 7.6e+07 | 8.3e+08 | 1.4e+08 | 1.2+e08 |
| feeding | 18.3 | 18.4 | 17.9 | 18.2 | 6.5 | 2.8 | 7.3 | 5.4 | 2.0e+08 | 3.1e+08 | 4.0e+08 | 2.3+e08 |
| 60–21 | | | | | | | | | | | | |
| baseline | 0.5 | 0.5 | 1.3 | 0.5 | 3.0 | 0.6 | 51.2 | 7.9 | 3.0e+08 | 7.2e+07 | 8.7e+08 | 6.7e+08 |
| feeding | 4.4 | 0.9 | 1.6 | 0.9 | 9.6 | 4.8 | 57.2 | 8.6 | 4.6e+08 | 1.4e+08 | 8.7e+08 | 4.1e+08 |
| 87–21 | | | | | | | | | | | | |
| baseline | 2.8 | 0.7 | 0.2 | 0.1 | 2.2 | 4.6 | 3.1 | 0.4 | 1.9e+08 | 9.1e+07 | 7.7e+07 | 2.0e+07 |
| feeding | 2.6 | 6.4 | 3.9 | 0.3 | 2.4 | 12.7 | 40.9 | 0.9 | 1.2e+08 | 4.7e+08 | 9.8e+08 | 9.5e+07 |
| 103–21 | | | | | | | | | | | | |
| baseline | 0.0 | 0.0 | 0.0 | 0.0 | 1.4 | 0.0 | 0.0 | 0.0 | 2.0e+08 | 1.5e+08 | 6.4e+07 | 4.9e+07 |
| eeding | 0.0 | 0.0 | 0.0 | 0.0 | 4.5 | 0.0 | 0.1 | 0.0 | 5.7e+08 | 2.0e+08 | 1.4e+08 | 7.4e+07 |

Red indicates channels with low quality based on the criterion of > 20% removal; Power amplitude is in E (scientific) notation

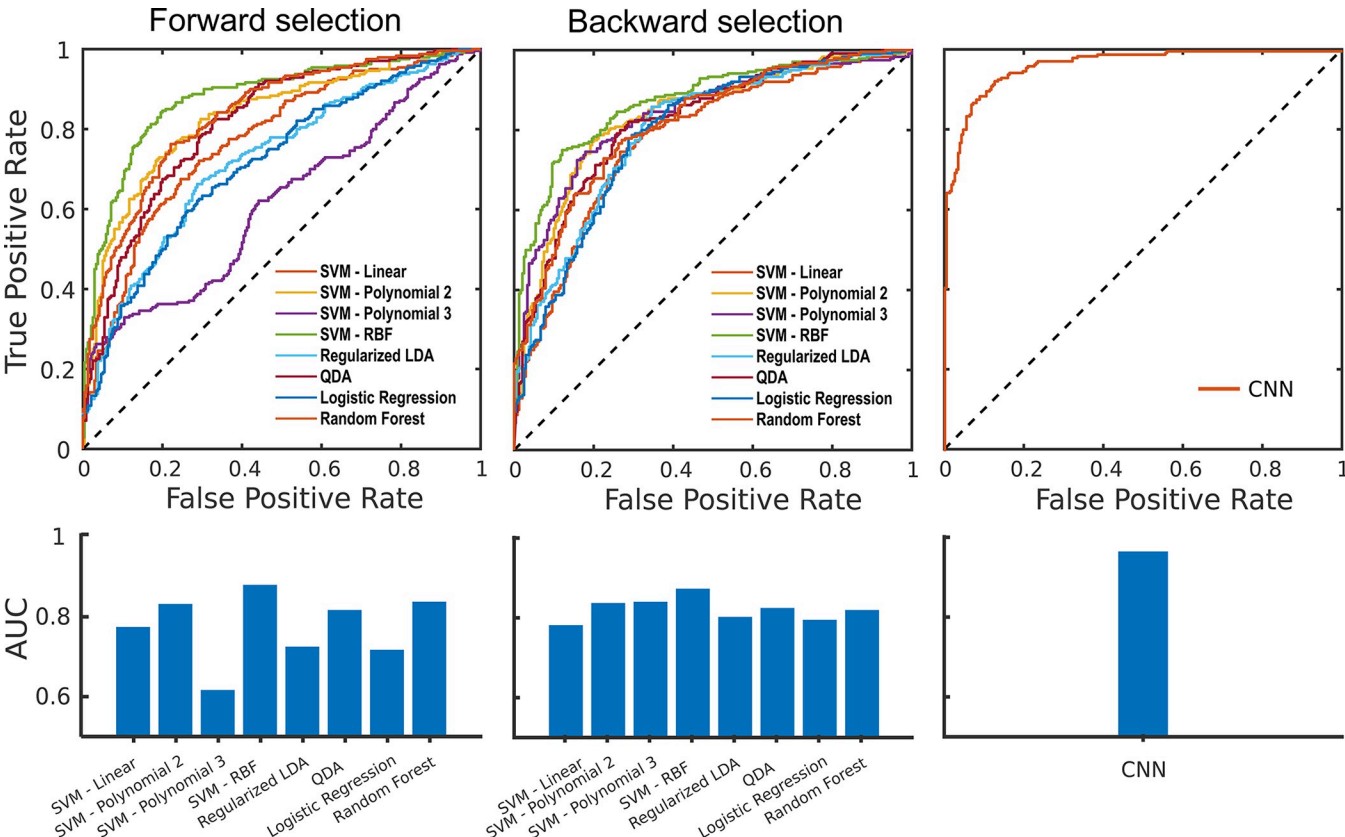

**Fig 4. Classifiers performance-ROC curves and AUC values.** A) ROC curves and AUC values for the frequency-based methods using forward feature selection. B) ROC curves and AUC values for the frequency-based methods using backward feature selection. C) ROC curve and AUC for the deep learning approach.

performance was consistent between all the animals for all the methods used (results are shown in S1 Table).

## Discussion

Our results showed the NN approach outperforms other classifiers in distinguishing baseline and feeding. A close second place performance occurred with the SVM classifier using a RBF kernel at 0.88 AUC compared to 0.97 AUC for the NN approach. Moreover, backwards feature selection performed better than forward feature selection. There are several strengths of the current study, which could have positive impacts on clinical applications. Unlike hand-crafted features, we used an agnostic approach for feature selection with deep learning methods; this could capture unknown signaling information between gastric states. We also carefully assessed the quality of signals and removed data that were likely non-physiological (i.e., signals above 2 mV). These artifacts likely occur in real world data acquisition of gastric myoelectric signals from patients equipped with internal or external electrodes, especially during movement. While this study incorporated a relatively small sample size of subjects (N = 4) and a short duration of recording, it could serve as a useful template to assess gastric function in individual patients with short test sessions.

Traditional machine learning approaches like support vector machines or discriminant analysis often use hand-crafted features, designed to capture relevant information in the data.

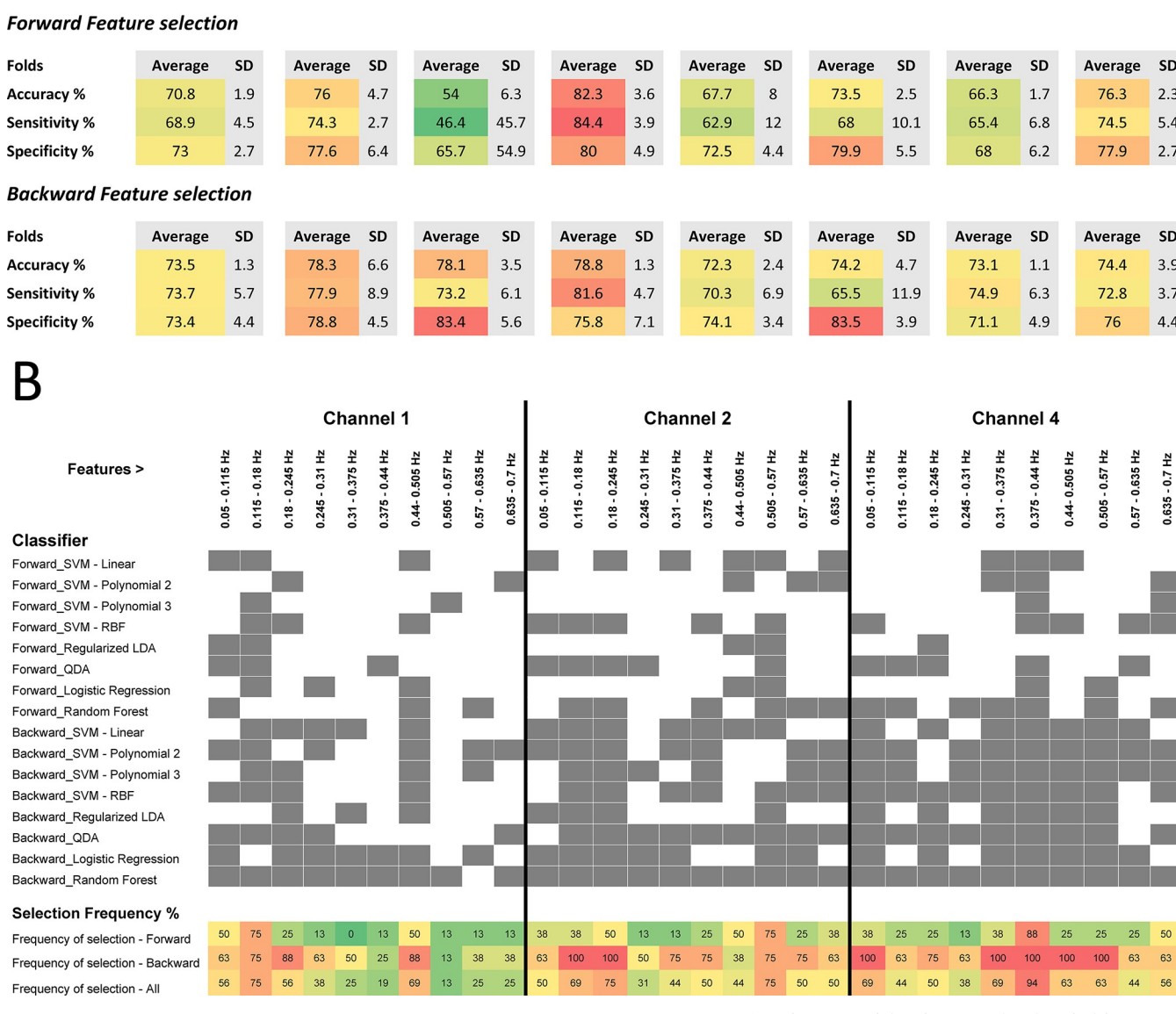

**Fig 5. Performance and feature selection of classifiers using features for each signal frequency.** A) Performance of classifiers using hand-crafted features. A heat map is used to indicate average performance across the folds for each classifier (red = high percentage to green = low percentage). SVM = Support Vector Machine; LDA = Linear discriminant analysis; QDA = Quadratic discriminant analysis. B) Contribution of power frequency features to classifier performance. Grey shading indicates that the feature was selected by the classifier. The bottom rows indicate the overall percentage that each feature was selected across all classifiers using a heat map (red = high percentage to green = low percentage).

In contrast, convolutional neural networks (e.g., the NN approach used in this paper) can learn to extract relevant features from the raw input signal by capturing complex temporal and spatial dependencies in the data, without being restricted to hand-crafted features. This could lead the neural network approach to outperform the other approaches. However, an important trade-off is reduced model interpretability and higher risk of bias and overfitting with the NN approach.

The limitations of the study include model complexity, use of invasive electrodes, and the gastric states used for testing. In general, the NN model provided no insight into what data features produced the model performance. It is also unclear if this classifier performance can be achieved using a less invasive approach to record signals, such as the typical surface electrodes used in clinical EGG. Finally, the states used here (i.e., baseline and feeding) are potentially extreme differences in physiological state and more subtle changes, such as reduced gastric function in gastroparesis patients compared to normal controls, might not be detectable.

Our results suggest several future investigations using preclinical models, followed by testing in humans. These studies could include the use of larger samples with more animals and potentially longer recording sessions under different testing conditions to determine whether the NN approach could be applied to predict gastric physiological states in individual animals. Importantly, this could be useful for personalized diagnostic medicine. Changes to the recording methods should also be investigated, including the use of fewer internal electrodes and dermal surface techniques coupled with the NN approach. In human studies, the non-invasive collection of data and subsequent NN approach could be readily investigated, although there may be limitations on the ability to record low-noise signals non-invasively during unconstrained behavior or in medical conditions such as gastroparesis, where signal amplitudes may be highly suppressed, necessitating implanted approaches [20]. Fortunately, recent evidence from our lab suggests that it should be possible to implant recording electrodes on the serosal surface of the stomach via minimally invasive surgical approaches [21].

## Supporting information

**S1 Table. Performance of classifiers per animal.** In the table, a heat map is used to highlight performance (red = high percentage to green = low percentage). Results were obtained with a detection threshold of 0.5. We show the average and standard deviation of the forward and backward methods (8 methods for each) to summarize them.
(DOCX)

## Acknowledgments

We thank Michael Sciullo for assistance with data collection and preliminary analysis.

## Author Contributions

**Conceptualization:** Murat Akcakaya, Stephanie Fulton, Charles C. Horn.

**Data curation:** Mahmoud Elkhadrawi, Charles C. Horn.

**Formal analysis:** Mahmoud Elkhadrawi, Charles C. Horn.

**Funding acquisition:** Bill J. Yates, Lee E. Fisher, Charles C. Horn.

**Investigation:** Charles C. Horn.

**Methodology:** Mahmoud Elkhadrawi, Murat Akcakaya, Charles C. Horn.

**Resources:** Bill J. Yates, Lee E. Fisher, Charles C. Horn.

**Software:** Mahmoud Elkhadrawi, Charles C. Horn.

**Supervision:** Murat Akcakaya, Stephanie Fulton, Bill J. Yates, Lee E. Fisher.

**Validation:** Mahmoud Elkhadrawi, Charles C. Horn.

**Visualization:** Mahmoud Elkhadrawi, Charles C. Horn.

**Writing – original draft:** Mahmoud Elkhadrawi, Murat Akcakaya, Charles C. Horn.

**Writing – review & editing:** Bill J. Yates, Lee E. Fisher.

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
