## [Decision Letter · Decision Letter 0]

8 Mar 2023

PONE-D-22-33411Prediction of gastrointestinal functional state based on myoelectric recordings utilizing a deep neural network architecturePLOS ONE

Dear Dr. Elkhadrawi,

Thank you for submitting your manuscript to PLOS ONE. After careful consideration, we feel that it has merit but does not fully meet PLOS ONE’s publication criteria as it currently stands. Therefore, we invite you to submit a revised version of the manuscript that addresses the points raised during the review process.

We look forward to receiving your revised manuscript.

Kind regards,

Zhishun Wang, Ph.D.

Academic Editor

PLOS ONE

Journal Requirements:

Reviewers' comments:

Reviewer's Responses to Questions

**Comments to the Author**

1. Is the manuscript technically sound, and do the data support the conclusions?

Reviewer #1: Yes

Reviewer #2: Partly

2. Has the statistical analysis been performed appropriately and rigorously? 

Reviewer #1: Yes

Reviewer #2: N/A

3. Have the authors made all data underlying the findings in their manuscript fully available?

Reviewer #1: Yes

Reviewer #2: Yes

4. Is the manuscript presented in an intelligible fashion and written in standard English?

Reviewer #1: Yes

Reviewer #2: Yes

5. Review Comments to the Author

Reviewer #1: Summary:

In this study, the authors aimed to classify gastric myoelectric signals into feeding and baseline control states using machine learning techniques. They compared the performance of a deep neural network approach with other traditional machine learning approaches that use hand-crafted features. The deep learning approach outperformed other classifiers, with an AUC value of 0.96, while SVM with RBF kernel achieved a close second place performance, with an AUC value of 0.88. The study's strengths include the use of an agnostic approach for feature selection with deep learning methods, careful signal quality assessment, and a relatively small sample size of subjects with short test sessions. However, the limitations included model complexity, the use of invasive electrodes, and potentially extreme physiological states used for testing. Future investigations could include the use of potentially longer recording sessions and different testing conditions to determine whether the NN approach could be applied to predict gastric physiological states in individual animals and humans.

The motivation behind this research is to develop an automated method for detecting gastric myoelectric activity and to compare the performance of machine learning approaches, including a deep neural network, with traditional approaches that use handcrafted features.

Strengths:

• The study proposes a new approach for gastric myoelectric signals analysis using deep learning, which shows better performance than traditional machine learning algorithms.

• The authors carefully assessed the quality of signals and removed data that were likely non-physiological.

• The study provides a useful template to assess gastric function in individual patients with short test sessions.

• The findings could have positive impacts on clinical applications in personalized diagnostic medicine.

Weaknesses:

• Small sample size: The study was conducted on a small sample size of only four ferrets, which limits the generalizability of the findings to other animal models or humans.

• Short duration of recordings: The study used only 1-hour recordings, which might not be sufficient to capture subtle changes in gastric myoelectric signals and their classification.

• Invasive electrode placement: The use of invasive electrodes in ferrets may not be applicable to humans, where non-invasive techniques, such as surface electrodes, are preferred.

• Limited gastric states used for testing: The study only tested the classification performance between feeding and baseline control states, which are potentially extreme differences in physiological state. More subtle changes, such as reduced gastric function in gastroparesis patients compared to normal controls, might not be detectable.

• Complexity of deep learning models: The deep learning models used in the study are complex and provide no insight into what data features produced the model performance. This lack of interpretability could be a limitation in clinical applications where interpretability and transparency are crucial.

Questions:

1. Can you provide more insight into the features that the deep neural network approach used for classification?

2. Have you considered investigating the performance of the deep learning approach on a larger sample size of subjects or recordings?

3. Is it possible to achieve similar performance using less invasive methods for signal acquisition, such as surface electrodes commonly used in electrogastrography?

4. How would you address the potential challenge of detecting more subtle changes in gastric function in gastroparesis patients compared to normal controls using the current approach?

Comments:

The authors should clarify some of the technical details of their approach, such as the specifics of the deep neural network architecture used, the hyperparameters selected, and the normalization method applied to the data.

The manuscript could benefit from additional proofreading to address any spelling or grammatical errors.

The authors should include more detailed information on the data collection process, such as the electrode placement and recording parameters used.

Reviewer #2: The presented method to detect the GI functional state based on myoelectrical recordings using deep neural networks. The approach is interesting however, the following points should be addressed.

1- The features used for training and predicting the output should be explained in detail i.e., frequency ranges of the 10 bands. Why were those bands used and how they differ theoretically during various GI functions?

2- False positives were not addressed in any case. Please explain how they effect the overall performance.

3-Please introduce all the ML approaches briefly and discuss their performance in more details. Why do you think NN outperformed others

4- The figure 5 is not legible, hence the numbers could not be verified. Please provide clear figure to verify the results.

5- Check the table number of the supplementary table

6- Please explain in more details about how the small signal duration can affect the results and how you solved with this issue.

6. PLOS authors have the option to publish the peer review history of their article (what does this mean?). If published, this will include your full peer review and any attached files.

Reviewer #1: No

Reviewer #2: **Yes: **M. Khawar Ali

---

## [Author Response · Author response to Decision Letter 0]

9 May 2023

Reviewer #1

Questions

1. Can you provide more insight into the features that the deep neural network approach used for classification?

We thank you for this question and we understand the interest for a deeper insight in the interpretability of neural network features. However, machine learning approaches are often referred to as “black box” models” due to the difficulty in understanding how they work. The goal of deep learning approaches is to capture complex non-linear patterns and relationships between the input and the output, but at the expense of a lack of understanding of those underlying relationships. In this paper a 1-minute signal is fed to a convolutional neural network to be classified by extracting non-trivial patterns within the raw signal. In conclusion, the trade-off for better model performance is having a more complex model with features that are difficult to interpret and explain. We describe this limitation in the third paragraph of the Discussion.

2. Have you considered investigating the performance of the deep learning approach on a larger sample size of subjects or recordings?

This is an excellent question, but we have not attempted to apply this approach to a larger cohort of animals. This will be the subject of future work. However, please note that even though the study was limited to four animals, we collected a large amount of data from each animal, which made it possible to apply deep learning methods. Moreover, in order to ensure that our approaches generalize across animals and do not overfit, all machine learning approaches were validated using 3-fold cross-validation. We have, however, added a statement in the fourth paragraph of the Discussion suggesting that future studies should include a larger sample size with more animals and longer recordings.

3. Is it possible to achieve similar performance using less invasive methods for signal acquisition, such as surface electrodes commonly used in electrogastrography?

This is an important question that we do not currently have data to address. However, there is good reason to believe that there would be major challenges in achieving the levels of performance reported here with non-invasive electrogastrogram recordings. As described by Yin and Chen (J Neurogastroenterol Motil, 2013), to achieve a low-noise reliable non-invasive electrograstrogram recording, patients “should be asked not to talk, move, read or make phone calls during the procedure.” In our study, animals were freely moving and active throughout the experiment. As such, while non-invasive approaches may be appropriate for diagnostic testing, an implanted approach may be necessary for applications that involve real-time classification of electrograstrogram signals under noisy conditions. We have added a statement at the end of the fourth paragraph of the Discussion addressing this issue.

4. How would you address the potential challenge of detecting more subtle changes in gastric function in gastroparesis patients compared to normal controls using the current approach?

As described above, we believe that there are multiple advantages to recording signals from implanted electrodes. For subtle changes in signals that might occur with patients with gastroparesis, we believe the low-noise capabilities of an implantable approach may be important, though we do not currently have evidence to confirm this hypothesis. We have added a statement at the end of the fourth paragraph of the Discussion, describing this potential advantage of an implanted recording system.

Comments:

The authors should clarify some of the technical details of their approach, such as the specifics of the deep neural network architecture used, the hyperparameters selected, and the normalization method applied to the data.

The manuscript could benefit from additional proofreading to address any spelling or grammatical errors.

The authors should include more detailed information on the data collection process, such as the electrode placement and recording parameters used.

We appreciate the suggestion regarding the addition of detail around the approach and have added a brief explanation about all the machine learning approaches used in this paper. These details are found in the “Machine learning” subsection (Methods and material> Data Analysis>Machine learning). The details of the convolutional neural network architecture, layers, hyperparameters, the kernels and optimizer are also provided.

We have also carefully proofread the paper and corrected grammatical mistakes where appropriate.

Details of the electrode placement are included in the second paragraph of the “Chronic electrode implant surgery” section of the Materials and Methods, and recording parameters are described at the end of the paragraph “Behavioral testing and gastric myoelectric recording” in the Material and Methods section. We have made edits to both sections to add additional clarity.

 

Reviewer #2

Questions

1- The features used for training and predicting the output should be explained in detail i.e., frequency ranges of the 10 bands. Why were those bands used and how they differ theoretically during various GI functions?

Slow wave activity in the ferret typically spans 3-18 cycles per minute (i.e., 0.05 to 0.3 Hz), and our recent work (Nanivadekar, et. al, 2019) has shown that interventions such as feeding or infusion of an emetic drug can increase signal power content at higher frequencies. As such, for machine learning approaches that used hand-crafted features, we chose to include a range of frequencies spanning those typically observed during behavior as well as higher frequencies (i.e. up to 0.7 Hz) to ensure we captured the frequencies most likely to change during feeding. Each of the 10 bands spanned 0.065 Hz to ensure sufficient resolution of individual features in the frequency domain to capture variability of different portions of the power spectrum. We have added additional detail to the sixth paragraph of the section titled “Machine Learning” in the Materials and Methods describing the justification for this choice of parameters.

2- False positives were not addressed in any case. Please explain how they effect the overall performance.

We investigated the overall performance using 3-fold cross-validation and showed the average sensitivity and specificity in Figure 5. Also, we plotted ROC curves for all the classification approaches in Figure 4. We noticed that the neural network (NN) approach achieved better results than the other approaches, thus the NN approach is more robust to false positives.

3-Please introduce all the ML approaches briefly and discuss their performance in more details. Why do you think NN outperformed others

Thank you for your recommendation. A brief introduction of all the applied machine learning methods was added to the machine learning subsection (Methods and material> Data Analysis>Machine learning). 

Traditional machine learning approaches like support vector machines or discriminant analysis often use hand-engineered features, designed to capture relevant information in the data. In contrast, convolutional neural networks (the NN approach used in this paper) can learn to extract relevant features from the raw input signal by capturing complex temporal and spatial dependencies in the data, without being restricted to engineered features. This could lead the neural network approach to outperform the other approaches. However, the trade-off is reduced model interpretability and higher risk of bias and overfitting.

4- The figure 5 is not legible, hence the numbers could not be verified. Please provide clear figure to verify the results.

A higher quality version of the figure was uploaded. Please let us know if it is not clear enough.

5- Check the table number of the supplementary table

This has been addressed in our resubmission. Thank you.

6- Please explain in more details about how the small signal duration can affect the results and how you solved with this issue.

We interpret this as a question about the size of the trials used for classification. Please note that we are not investigating changes in sample signal duration. We have collected a large amount of data (1 hour with sampling frequency 2000 Hz) from each animal, for both baseline and feeding conditions. We segmented those signals into smaller 1-minute chunks to generate multiple samples for each condition, which enabled us to train the CNN model. Then, we compared feeding and baseline conditions based on these observations through ML. This doesn’t mean that we investigate the changes across conditions based on small signal duration.

---

## [Decision Letter · Decision Letter 1]

11 Jul 2023

Prediction of gastrointestinal functional state based on myoelectric recordings utilizing a deep neural network architecture

PONE-D-22-33411R1

Dear Dr. Elkhadrawi,

We’re pleased to inform you that your manuscript has been judged scientifically suitable for publication and will be formally accepted for publication once it meets all outstanding technical requirements.

Kind regards,

Zhishun Wang, Ph.D.

Academic Editor

PLOS ONE

Additional Editor Comments (optional):

Reviewers' comments:

Reviewer's Responses to Questions

**Comments to the Author**

1. If the authors have adequately addressed your comments raised in a previous round of review and you feel that this manuscript is now acceptable for publication, you may indicate that here to bypass the “Comments to the Author” section, enter your conflict of interest statement in the “Confidential to Editor” section, and submit your "Accept" recommendation.

Reviewer #1: All comments have been addressed

Reviewer #2: All comments have been addressed

2. Is the manuscript technically sound, and do the data support the conclusions?

Reviewer #1: Partly

Reviewer #2: Yes

3. Has the statistical analysis been performed appropriately and rigorously? 

Reviewer #1: Yes

Reviewer #2: N/A

4. Have the authors made all data underlying the findings in their manuscript fully available?

Reviewer #1: Yes

Reviewer #2: Yes

5. Is the manuscript presented in an intelligible fashion and written in standard English?

Reviewer #1: Yes

Reviewer #2: Yes

6. Review Comments to the Author

Reviewer #1: 1. I understand the 'black-box' nature of your model. However, I am asking for the details of using the model. Consider exploring interpretability techniques in future

2. I appreciate the idea to test a larger cohort in future work. Expanding to other species could also be an interesting addition.

3. Your thoughts on non-invasive approaches are insightful. Despite challenges, pursuing such methods would be a valuable direction for future studies.

4. Investigating whether implanted electrodes better detect subtle gastroparesis changes is a solid plan. Empirical evidence will be key.

Your clarifications on the deep learning approach, grammar corrections, and details on electrode placement and recording are helpful.

Reviewer #2: (No Response)

7. PLOS authors have the option to publish the peer review history of their article (what does this mean?). If published, this will include your full peer review and any attached files.

Reviewer #1: No

Reviewer #2: **Yes: **M. Khawar Ali

---

## [Editor Report · Acceptance letter]

19 Jul 2023

PONE-D-22-33411R1 

Prediction of gastrointestinal functional state based on myoelectric recordings utilizing a deep neural network architecture 

Dear Dr. Elkhadrawi:

I'm pleased to inform you that your manuscript has been deemed suitable for publication in PLOS ONE. Congratulations! Your manuscript is now with our production department. 

Kind regards, 

on behalf of

Dr. Zhishun Wang 

Academic Editor

PLOS ONE